# Working Conditions and Sick Building Syndrome among Health Care Workers in Vietnam

**DOI:** 10.3390/ijerph17103635

**Published:** 2020-05-21

**Authors:** Cuong Hoang Quoc, Giang Vu Huong, Hai Nguyen Duc

**Affiliations:** 1Pasteur Institute, Ho Chi Minh City 700000, Vietnam; haiyds@gmail.com; 2Public Health Faculty, Hong Bang Medical Center, Hai Phong 180000, Vietnam; chizuluvnaoki@gmail.com

**Keywords:** sick building syndromes, health care workers, Vietnam

## Abstract

Background: Little is known about risk factors for sick building symptoms (SBS) among health care workers (HCWs) who often face the workload, exposure to chemicals, and biological contaminants in the workplace. This study aims to evaluate the correlation between SBS and the symptoms among HCWs. Methods: A total of 207 HCWs were recruited in a large hospital-based cross-sectional survey between March and June 2017, southern Vietnam. Face-to-face interviews were conducted for collecting data on demographics, SBS-related symptoms, working environments, and conditions. Indoor environmental conditions were measured. SBS scores, ranging from 0 to 24, were determined by a sum of the scores of general symptoms, mucosal irritation, and skin symptoms; multivariate regression analyses and the Lindeman, Merenda, and Gold (LMG) test were used to investigate the predictors and its impact on the SBS. Results: A mean SBS score was 9.7 (range: 1–21). Compared with males, females were more likely to report higher SBS scores (10.2 vs. 7.9, *p* < 0.001). Being female, atopy, varying temperature room, stuffy “bad” air dust, and dirt had higher SBS scores of 2.0; 1.8; 1.7; 1.9; 3.8, respectively. LMG test showed that dust and dirt, and stuffy “bad” air were the predominant risk factors for SBS. Conclusions: Our study reveals that working conditions are important and significantly associated with SBS. Taken together with our findings, the working condition criteria approach trained for architects, builders, owners, and maintenance of the building is highly recommended for indoor air quality improvement. Furthermore, larger-sample studies about working condition are urgently needed to better manage SBS.

## 1. Introduction

Sick building syndrome (SBS) is typically manifested as irritations of the skin and mucous membranes and other symptoms, including headache, fatigue, and difficulty concentrating, which predominantly affects workers in modern office buildings [1,2].

Up till now, SBS has just identified based on the exclusion of other detectable illness, therefore, it causes emerging health risk concerns due to reducing work efficiency, increasing absenteeism and tardiness, and raising the healthcare budget [3,4,5].

Workers spend approximately 90 percent time indoors for work, unfortunately, the indoor air condition is sometimes more polluted than outdoor [4]. Additionally, urbanization and land shortage accompanied by densely populated conurbation have led to this situation getting worse as inadequate air ventilation and pollution occur inside high-rise buildings in large cities [6,7,8,9,10]. Several previous studies conducted on health care workers (HCWs) showed approximately 84% of participants suffered from at least one SBS symptom in China [11]; while in Iran and Turkey, the prevalence of SBS was 86.4%, 20.9%, respectively [6,12].

SBS is associated with multiple factors, including poor ventilation, outdoor and indoor chemicals, biological contaminants, females, allergy, smoking, workload, with less social support reported around the world such as China, Denmark, United Kingdom, Ethiopia, Sweden, Iran, Turkey; [12,13,14,15,16,17,18,19]. In particular cases, high CO_2_ levels related to symptoms like nausea, headaches, nasal irritation, dyspnea, and throat dryness, high light intensity were likely to report skin dryness, eye pain, and malaise. Similarly, high temperatures correlated with symptoms such as sneezing, skin redness, itchy eyes, and headache, while high relative humidity was likely to report sneezing, skin redness, and pain of the eyes [20]. In addition, workload and low work satisfaction associated with general symptoms (headache, abnormal tiredness, a sensation of cold or nausea) and upper respiratory symptoms [21]. Lower respiratory symptoms were related with high workload, longer work hours, chemical exposure, migraine, and exposure to new interior painting [22].

There is research on SBS on workers such as officers, HWCs, however, SBS among HCWs in Asia is still an interesting topic, and this study is the first research to assess the SBS-related risk factors in Vietnam. Furthermore, little is known about risk factors for SBS-related symptoms among HCWs who must be good physical and mental health conditions as HCWs play the crucial responsibilities in social life for taking care of patient’s health. This study aims to evaluate the correlation between SBS and the symptoms among HCWs.

## 2. Material and Methods

This cross-sectional study was conducted at the University Medical Center at Ho Chi Minh City. The study protocol was reviewed and approved by the institutional review board at the Medicine and Pharmacy at Ho Chi Minh City (HCMC), Vietnam (reference number: 137/ĐHYD-HĐ). All participants completed the informed consent form before the interviews.

### 2.1. Study Population and Study Design

The target population of the present study was Vietnamese HCWs aged over 18 years old and experienced equal or more than 8 h per day at a study setting. The standard deviation of SBS score was estimated at 5.4 [11], and the desired precision was set at 5%, indicating that 113 HCWs were needed; the design effect (average 26 HCWs per each department) was 1.8, the sample size was rounded to 207 participants, allowing 5% for incomplete data.

The present study was conducted between March and June 2017 at the University Medical Center HCMC, Vietnam; which is one of the prestigious hospitals in Vietnam, with 15 floors and approximately 2279 workers.

A two-stage recruitment strategy was used in this study. First, based on the conveyed-on department size of hospital offices, venues for recruitment were randomly selected, and we then selected eight out of 53 departments. In each department, we recruited HCWs who worked 8 h per day and experienced over 3 months in the selected departments. Second, based on the list of HWCs in selected departments, we then approached and invited HCWs to take part in this survey.

### 2.2. Data Collection

HCWs were informed of the study purposes, contents, and interview methods, given answer sheets and informed consent. The training interview then administered a standardized questionnaire to collect quantitative information about respondents. We used the NMO40A questionnaire to collect data that were translated into Vietnamese based on a pilot search on 20 HCWs [18]. The reliability and validity of the working environment and SBS symptoms were confirmed by Cronbach’s alpha test, which were 0.85 and 0.82, respectively. The study interview schedule included 34 questions separated into four domains: demographics, working environment, working condition, SBS symptoms over the previous 3 months. Face-to-face interviews were conducted in a private room in each department.

### 2.3. Demographics

In this study, we collected information on sex, age, smoking status, atopy (defined as having asthma or hay fever), occupation, and working experiences.

### 2.4. Working Environment

A number of factors related to working environments were collected, such as draught, room temperature (too high, varying, too low), stuffy “bad” air, unpleasant odor, static electricity, passive smoking, noise, light that is dim or causes glare and/or reflections, dust, dirt, and being bothered in work during the previous three months.

### 2.5. Working Conditions

The working conditions were determined by each participant who answered questions about being bothered during the last three months related to twelve physical indoor climate factors. There were four questions measuring working conditions as work satisfaction (regarding his or her work as interesting and stimulating), workload (having too much work to do), social support (having fellow workers help with problems in work), and job control (having any opportunity to influence his or her working condition) [23].

### 2.6. SBS Related Symptom Score (SBS Score)

SBS related symptoms were categorized as “yes, of often (≥2 times/week)” = two, “yes, sometimes (1 time/week)” = one, and “no, never” = zero. SBS score range is between 0 and 24. It means that the higher SBS score is, the more serious the level of SBS. In detail, five questions on general symptoms (fatigue; feeling heavy-headed; headache; nausea or dizziness; and difficulty concentrating), four questions on mucosal irritation (itching, burning, or irritation of the eyes; irritated, stuffy or runny nose; hoarse, dry throat; and cough), and three on skin symptoms (dry or flushed facial skin; scaling or itching scalp or ears; and dry, itching, red skin) [11,24].

Before study implementation, a two-day training course on the study procedures, study forms, and data collection was provided to the study interviewers. The interviewers then undertook a pilot interview to check the translation and comprehension of the interview questions, practice the consent process, using written notes. The pilot interview was conducted with any necessary changes to the guide. It took approximately twenty minutes to interview each participant. All written notes of the interview were kept in the possession of the interviewer while in the field.

### 2.7. Environmental Monitoring

In this study, we measured air quality in each selected department. Room temperature and relative humidity were measured with a Rotronic RH Temp Sensors (HF420, Bassersdorf, Switzerland), lighting (SIBATA F11, Soka, Saitama, Japan); electromagnetic radiation (Chauvin Arnoux C.A 43,Vire, Normady, France), level of noise (RION NA27, Osaka, Japan), air velocity (Kanomax 6036-CE, Osaka, Japan).

CO_2_ concentration was monitored continuously for 1 h using an AQ 200 air quality (Chevry-Cossigny, Seine-et-Marne, France). The levels of PM were monitored continuously for 1 h using aDustTrak™ II Aerosol Monitor 8532 (Shoreview, MN, USA). These indicators were also recorded during sampling, the measurements were performed in the morning of the interview day. All equipment was calibrated before implementation.

### 2.8. Data Analysis

All interview answer sheets had been checked by the interviewers for any missing information before the interview ended and then stored in locked cabinets.

Data were entered using Epi-Data version 3.1 (EpiData Association, Odense, Denmark, 2005), and all statistical analyses were carried out using Stata version 13.0 (StataCorp. 2013. Stata Statistical Software: Release 13. College Station, TX: StataCorp LP), in R version 3.4.1 (R Core Team, Vienna, Austria, 2014), and *p*-value < 0.05 was considered statistically significant.

The sample’s characteristics were summarized using frequency and proportion for categorical variables, and mean and stand deviation or median and interquartile range for continuous variables.

Continuous variables were compared using Student’s *t*-test. Linear regression analysis was performed using SBS score. In this process, if continuous variables yielded a *p*-value below 0.05 or were previously known to be an important risk factor (e.g., sex, or smoke, allergy), these were included in a multivariable model [25,26]. In multivariable analysis, we used the Bayesian model average (BMA); Bayesian information criterion (BIC) used to verify whether the final model was indeed the optimal model. Of the 2^n^ models (n was the number of independent variables), the optimal model was selected if it has a lower number of independent variables, a lower BIC, a higher R-squared, and a higher post probability [27].

Lastly, the impact of each risk factor on SBS scores was assessed by the Lindeman, Merenda, and Gold (LMG) test which corresponded to a *p* < 0.05 [28].

## 3. Results and Discussion

### 3.1. Demographics

From March to June 2017, a total of 207 HCWs were recruited. The median age and working experience of participants were 27 years old (interquartile range, 25–32) and they had three years of working experience (interquartile range, 0.5–15), respectively. Over one-half of the participants (54.6%) were nurses, less than one in ten smoked. Suffering from atopy was reported in 57% of HCWs.

### 3.2. Working Environment

In terms of the working environment, 30.4%, 25.6%, and 14.5% of HCWs complained of varying room temperature, noise, the unpleasant odor, respectively.

### 3.3. Working Conditions

We found that 97.1% of HCWs reported workload, a higher proportion of HCWs reported work satisfaction (87.9%), social support, and job control (60.9%).

### 3.4. SBS Score

The five predominant weekly complaints about SBS symptoms were fatigue (98.6%), feeling heavy-headed (89.4%), headache (89.9%), cough (83.6%), and hoarse/dry throat (74.4%).

In this study, the mean SBS score was 9.7 ± 3.9 (range, 1–21); in detail, SBS-score for male and female HCWs was 7.9 ± 4.0; 10.2 ± 3.7, respectively.

Several significant differences in demographic characteristics were observed between male and female HCWs. Compared with male HCWs, female HCWs had significantly lower median age (27 vs. 31), were more likely to be nursing (64.4% vs. 21.2%), less likely to be smoking (0.6% vs. 14.9%). Female HCWs had higher SBS scores (10.2 vs. 7.9), feeling heavy-headed (91.9% vs. 80.9%), headache (93% vs. 78.7%), hoarse/dry throat (78.1% vs. 61.7%), cough (85.6% vs. 76.6%), dry or flushed facial skin (64.4%, 42.2%), hand dry/itching/redskin (61.9% vs. 31.9%) (Table 1).

### 3.5. Indoor Environmental Conditions, the Association between SBS Score and Chemical Concentrations

During the collecting sampling period, the mean air temperature, relative humidity, and CO_2_ were 26.4 ± 1.2 °C (ranged from 23 to 28), 60.6 ± 5.4 (ranged from 57 to 74), 1720 ± 299.4 (ranged from 1430 to 1875), respectively.

In this study, the indoor environmental conditions did not explain the SBS change among health workers (*p* > 0.05), except for the lighting factor. When the lighting increased 1 lux, the SBS score increased by 0.01 points (*p* = 0.05), and it explained approximately 51.2% of the SBS change (Table 2).

### 3.6. Association between SBS Score and Characteristics, Working Environment, Working Condition

In univariate analysis, female HCWs (ß = 2.33; 95% Cl: 1.10, 3.57), no smoking (ß = 3.19; 95% Cl: 0.46, 5.93), atopy (ß = 2.23; 95% Cl: 1.19, 3.26) had a significantly higher SBS score. Being technicians (ß = −3.14, 95%Cl: −5.90, −0.38) and pharmacists (ß = −5.47; 95%Cl: −9.89, −0.38) had lower SBS scores (Table 3).

HCWs who reported complaining of draught (ß = 2.40, 95% Cl = −0.35–5.15), varying room temperature >5 °C (ß = 2.29, 95% Cl = 1.17–3.40), stuffy “bad” air (ß = 3.54, 95% Cl = 1.47–5.61), dry air (ß = 2.94, 95% Cl = 1.18–4.70), unpleasant odor (ß = 2.15, 95% Cl = 0.66–3.64), noise (ß = 3.07, 95% Cl = 1.92–4.22), light (ß = 3.44, 95% Cl = 1.77–5.11), dust and dirt (ß = 3.85, 95% Cl = 1.92–5.78) had a significantly higher SBS score.

Higher likelihoods of SBS score were observed among HCWs reporting workload (ß = 5.30, 95% Cl = 2.08–8.28), and job control (ß = 1.91, 95% Cl = 0.85–2.98).

In multivariable analysis, significantly higher SBS scores were observed in HCWs who were female (2.0; 95%Cl: 0.91, 3.13), had atopy (1.8; 95%Cl: 0.85, 2.74), and complained of varying room temperature (1.7; 95%Cl: 0.70, 2.73), stuffy “bad” air (1.9; 95%Cl: 0.01, 3.78), and dust and dirt (3.8; 95%Cl: 2.05, 5.54).

The optimal model with R-square of 0.43, BIC of −89.2, and post-probability of 0.12 had an equation as per following:

SBS = 4.2 + 2.0 (female) + 1.7 (varying temperature room over 5 °C) + 1.9 (stuffy “bad” air) + 3.8 (dust and dirt) + 1.8 (atopy) (Table 4).

Three-domain variables that had the highest impacts on SBS scores were dust and dirt (LMG = 0.10), stuffy “bad” air and light (LMG = 0.70), while workload factor had the lowest impact on SBS scores (LMG = 0.02) (Table 5).

In this study of 207 HCWs in a big hospital located in southern, Vietnam, we found that HCWs were more likely to report the risk factors associated with SBS and a high level of SBS symptoms in the previous 3 months. In our study, over 90% of participants suffered from general symptoms like fatigue, feeling heavy-headed, and headache, which were higher than previous studies. In detail, a study conducted on HCWs in China showed that just 30% reported weekly complaints of fatigue; 19% reported headache [11]; among nurses in Iran, headache (83.3%) and fatigue (89.6%) [6]; while among HCWs in Sweden, fatigue (30%) was the most common general symptoms of SBS [28]. Furthermore, the estimate SBS score of HCWs is considerably higher than the average level reported in London with a SBS score of 2.2 for male and 2.7 for females, respectively [25]. In our study, we also found that 38% of males (18/47) and 60% of females (96/160) had a SBS score equal and more than an average SBS score (9.7). These data suggest that a SBS score and the severity of SBS were fairly high among HWCs. Of concern, varying room temperature, stuffy “bad” air, dust and dirt, light were significant associations with a mean SBS score among HCWs. This may consequently prolong SBS symptoms, which cause low productivity at work, increasing absences at work, and raising health care spending [11].

Most indoor environmental conditions reached the accepted criterion of the Vietnam regulation [29]; however, temperature, noise, and lighting are not in agreement with the previous study [30]. It could be explained that these criteria depended on the climate area in the study settings.

CO_2_ was a criterion to determine ventilation efficacy [11,24]. In this study, we found that the concentration of CO_2_ exceeded 1500 ppm, which was higher than previous studies [11]. The CO_2_ level was high, which shows that the hospital was not well ventilated as air velocity (0.3 ± 0.1) was quite low and unclearly a little above the recommended level of 0.2–2 m/s, as suggested by the Vietnam Ministry of Natural Resources and Environment [29]. However, our study is not agreement with findings showing that CO_2_ concentration was a significant negative association with the SBS score. It could be explained in this study due to small sample sizes.

Lighting was higher than the recommended level of 300 lux by Vietnam regulation [29]. Our study found that lighting related to the SBS score, which was in agreement with earlier findings [30]. In particular, the mean level of lighting was lower than the previous study, where the recommended level of lighting was 500 lux for general office work [31]. It could be explained that lighting has the potential to affect general satisfaction in workplace and comfort of visual performance.

In the present study, we found that the varying room temperature was more likely to suffer from SBS, which was consistent with the previous study depicting that the variation of room temperature 22–26 °C and the temperature over 23 °C could increase the sensation of dryness and mucosal irritation symptoms [19,32]. It is possible that the setting study is located in the tropical climate area; and we collected data in the summer season when HCWs usually use the air conditioner, resulting in a variation in temperature between outdoor and indoor in the workplaces. Another thing is that, as a big hospital with interdisciplinary, HCWs shifted from one department to another, which also contributed to the varying temperature between indoor and outdoor in their workplace, therefore, the varying room temperature (30.4%) in our study was higher than a study in Malaysia (11.5%) [24].

HCWs complaints of stuffy “bad” air in the workplace was associated with SBS [12]. Although, PM_10_ was within the levels recommended in Vietnam regulation [29], the present study also showed that the dust and dirt factor inside the hospital was more likely to be reported SBS [12]. It could be explained that stuffy “bad” air, and dust and dirt factors combined with chemicals and biological contaminants in the workplace contribute to increasing SBS symptoms, especially mucosal irritation and skin symptoms [6,12]. Possible reasons include many patients inside the study setting, high humid weather leading to the low quality of ventilation surrounding their workplace [6,12]. It may be explained that the hospital sector in our study was built near the main roads where too many private vehicles surround the hospital sector and we did not measure the level of PM_2.5_.

Moreover, we found that participants affected by atopy had an association with SBS, which was consistent with the previous studies among hospital workers, office workers, and the general population [33,34,35].

Based on the results of this study, we also found female HCWs were associated with a much higher likelihood of SBS, in agreement with the previous study [36,37], however, in some cases reported, SBS had no difference in sex among HCWs [6,38]. This was more likely to be more exacerbated as female HCWs accounted for the majority of the workforce in the hospital environment; in our study showed that over 77% of females, compared with 84.2%, 82.5%, 61.6% in Iran, China, and Turkey, respectively; furthermore, most of them complained SBS symptoms and had high SBS score [6,11,12].

Although we found there were risk factors associated with SBS, our findings should be approached with limitations. SBS symptoms, working conditions, working environments were accessed in this study, so we could underestimate the correlates of other risk factors for SBS among HCWs as bacteria, fungi, carbon monoxide, total volatile organic compounds, formaldehyde, and ozone, PM_2.5_, which were reported in previous studies [11,19,24,39]. In terms of working condition, when we translated into Vietnamese and accessed the reliability and validity of the working environment confirmed by Cronbach’s alpha test were 0.45; underreporting these answers related to working condition and working environment could occur during face-to-face interviews as HCWs were always busy during the study time and limited our efforts to examine the predictors of SBS. However, we contacted participants first and only interviewed on the weekend or once respondents had free time.

## 4. Conclusions

HCWs in hospitals are exposed to the physical stressors in working conditions which have a potential increased SBS score and severity of SBS. The risk factors could be varying room temperature, dust, and dirt, lightning, and stuffy “bad” air were the predominant influences for SBS. Our study reveals that working conditions are important and significantly associated with SBS. Taken together with our findings, the working condition criteria approach trained for architects, builders, owners, and maintenance of the building is highly recommended for indoor air quality improvement. Furthermore, larger sample studies about working conditions are urgently needed to better manage SBS.

## Figures and Tables

**Table 1 ijerph-17-03635-t001:** Sick building syndrome (SBS) symptoms, working environment, and working condition characteristics among female and male healthcare workers (n = 207).

Variables	Female		Male		Total		*P* *
	N	%	N	%	n	%	
**Characteristic**							
Age (yr), median (IQR)	160	27 (22–40)	47	31 (25–43)	207	27 (22–50)	<0.001
Occupation							
Nursing	103	64.4	10	21.2	113	54.6	<0.001
Doctors	12	7.5	21	44.7	33	15.9	
Technicians	2	1.3	6	12.8	8	3.9	
Pharmacists	1	0.6	2	4.3	3	1.4	
Aides	27	16.9	3	6.4	30	14.5	
Others	15	9.3	5	10.6	20	9.7	
Atopy §	97	60.6	21	44.7	118	57.0	0.052
Working experience (yr), median (IQR)	160	3 (2.9–4.3)	47	4 (2–6)	207	3 (2.8–5)	0.185
Smoke	1	0.6	7	14.9	8	3.9	<0.001
**Sick Building Syndrome (SBS)**							
SBS score, mean (SD)	207	10.2 ± 3.7	207	7.9 ± 4.0	207	9.7 ± 3.9	<0.001
Fatigue	159	99.4	45	95.7	204	98.6	0.130
Feeling heavy-headed	147	91.9	38	80.9	185	89.4	0.031
Headache	149	93	37	78.7	186	89.9	0.004
Nausea/dizziness	97	60.6	21	44.7	118	57.0	0.052
Difficulties concentrating	118	73.8	29	61.7	147	71.0	0.109
Itching/burning or irritation of the eyes	97	60.6	26	55.3	123	59.4	0.515
Irritated/stuffy or runny nose	118	73.8	28	59.6	146	70.5	0.061
Hoarse/dry throat	125	78.1	29	61.7	154	74.4	0.023
Cough	137	85.6	36	76.6	173	83.6	0.142
Dry or flushed facial skin	103	64.4	20	42.6	123	59.4	0.007
Scaling/itching scalp or ears	61	38.1	14	29.8	75	36.2	0.296
Hand dry/itching/redskin	99	61.9	15	31.9	114	55.1	<0.001
**Working environment**							
Draught	7	4.4	1	2.1	8	3.9	0.686
Room temperature too high	6	3.8	0	0.0	6	2.9	0.340
Varying room temperature	51	31.9	12	25.5	63	30.4	0.406
Room temperature too low	5	3.1	3	6.4	8	3.9	0.385
Stuffy “bad” air	12	7.5	2	4.3	14	6.8	0.741
Dry air	16	10.0	4	8.5	20	9.7	1.000
Unpleasant odor	24	15.0	6	12.8	30	14.5	0.702
Static electricity	5	3.1	0	0.0	5	2.4	0.590
Passive smoking	6	3.8	1	2.1	7	3.4	1.000
Noise	11	23.4	42	26.3	53	25.6	0.694
Light (dim or causes glare and/or reflections)	16	10.0	6	12.8	22	10.6	0.589
Dust and dirt	11	6.9	5	10.6	16	7.7	0.368
**Working condition**							
Work satisfaction	143	89.4	39	83.0	182	87.9	0.237
Workload	156	97.5	45	95.7	201	97.1	0.528
Job control	96	60.0	30	63.8	126	60.9	0.636
Social support	154	96.3	43	91.5	197	95.2	0.181

* *P* from Chi-square or Fisher exact tests for categorical variables and Student’s *T*-test for continuous variables. § Atopy was defined as asthma or hay fever.

**Table 2 ijerph-17-03635-t002:** Relationship between indoor environmental conditions and SBS score.

Variables	SBS (Mean ± SD)	ß	*p*	Cl 95%	R^2^
Temperature (°C)	26.4 ± 1.2	−0.60	0.50	−2.61–1.42	0.080
Relative humidity (%)	60.6 ± 5.4	−0.01	0.96	−0.5–−0.47	0.001
CO_2_ (mg/m^3^)	1720 ± 299.4	−0.005	0.19	−0.01–−0.003	0.269
PM_10_ (mg/m^3^)	5.1 ± 0.1	−21.0	0.06	−42.5–0.61	0.485
Air velocity (m/s)	0.3 ± 0.1	14.8	0.30	−16.84–46.52	0.180
Electromagnetic radiation (V/m)	0.5 ± 0.3	−3.4	0.46	−12.94–6.21	0.110
Lighting (lux)	356.9 ± 136.1	0.01	0.05	0.0003–0.03	0.512
Level of noise (dBA)	61.9 ± 7.8	−0.12	0.40	−0.43–0.20	0.121

**Table 3 ijerph-17-03635-t003:** Univariate analyses of correlates of SBS score among healthcare workers (n = 207).

Variables	SBS Score Mean ± SD	Unadjusted ß (95% Cl)	*P* *
**Sex**			
Male	7.9 ± 4.0	Reference	
Female	10.2 ± 3.7	2.33 (1.10–3.57)	<0.001
**Smoking**			
Yes	6.6 ± 4.0	Reference	
No	9.8 ± 3.8	3.19 (0.46–5.93)	0.02
**Atopy**			
No	8.4 ± 3.6	Reference	
Yes	10.7 ± 4.0	2.23 (1.19–3.26)	<0.001
**Age group**			
30	9.9 ± 3.8	Reference	
31-40	8.9 ± 4.0	−1.00 (−2.21–0.22)	0.11
>40	11.4 ± 3.9	1.52 (−1.43–4.48)	0.31
**Occupation**			
Nursing	10.1 ± 3.8	Reference	
Doctors	9.1 ± 4.0	−1.08 (−2.57–0.41)	0.16
Technicians	7 ± 3.7	−3.14 (−5.90–−0.38)	0.03
Pharmacists	4.7 ± 1.2	−5.47 (−9.89–−1.06)	0.02
Aides	9.8 ± 3.0	−0.37 (−1.93–1.18)	0.63
Others	10.0 ± 4.6	−0.19 (−2.02–1.64)	0.84
**Working experience**	---	−0.04 (−0.21–0.14)	0.67
**Draught**			
No	9.6 ± 3.9	Reference	
Yes	12 ± 3.1	2.40 (−0.35–5.15)	0.09
**Room temperature too high**			
No	9.7 ± 3.9	Reference	
Yes	11 ± 2.2	1.34 (−1.84–4.52)	0.41
**Varying room temperature**			
No	9 ± 3.9	Reference	
Yes	11.3 ± 3.3	2.29 (1.17–3.40)	<0.001
**Room temperature too low**			
No	9.7 ± 4.0	Reference	
Yes	10.0 ± 1.5	0.32 (−2.45–3.09)	0.82
**Stuffy “bad” air**			
No	9.5 ± 3.8	Reference	
Yes	13 ± 4.2	3.54 (1.47–5.61)	0.001
**Dry air**			
No	9.4 ± 3.8	Reference	
Yes	12.4 ± 3.5	2.94 (1.18–4.70)	0.001
**Unpleasant odor**			
No	9.4 ± 3.8	Reference	
Yes	11.5 ± 4.1	2.15 (0.66–3.64)	0.01
**Static electricity**			
No	9.6 ± 3.9	Reference	
Yes	12.2 ± 2.8	2.57 (−0.90–−6.03)	0.15
**Passive smocking**			
No	9.6 ± 3.9	Reference	
Yes	11.4 ± 4.8	1.80 (−1.15–4.74)	0.23
**Noise**			
No	8.9 ± 3.7	Reference	
Yes	12.0 ± 3.5	3.07 (1.92–4.22)	<0.001
**Light** (dim or causes glare and/or reflections)			
No	9.3 ± 3.8	Reference	
Yes	12.8 ± 3.1	3.44 (1.77–5.11)	<0.001
**Dust and dirt**			
No	9.4 ± 3.8	Reference	
Yes	13.3 ± 3.1	3.85 (1.92–5.78)	<0.001
**Work satisfaction**			
No	9.2 ± 4.9	Reference	
Yes	9.8 ± 3.7	0.61 (−1.03–2.25)	0.46
**Workload**			
No	4.7 ± 3.9	Reference	
Yes	9.8 ± 3.8	5.20 (2.08–8.28)	0.001
**Job control**			
No	8.5 ± 3.6	Reference	
Yes	10.4 ± 3.9	1.91 (0.85–2.98)	<0.001
**Social support**			
No	7.4 ± 5.3	Reference	
Yes	9.8 ± 3.8	2.41 (−0.06–4.89)	0.06

* *P* from Student’s *T*-test for continuous variables.

**Table 4 ijerph-17-03635-t004:** Four optimal models were chosen through the Bayesian model average method.

Equitation	No. Variables	R-Square	Post-Probability	Bayesian Information Criteria
**SBS score** = 4.2 + 2.0; Sex + 1.7 varying room temperature + 1.9; stuffy “bad” air + 3.8; dust and dirt + 1.8; atopy	5	0.43	0.12	−89.2
**SBS score** = 4.1 + 2.0; Sex + 1.6 varying room temperature + 1.5; stuffy “bad” air + 3.6; dust and dirt + 1.8; atopy + 1.4; dry air	6	0.44	0.11	−89.1
**SBS score** = 4.0 + 2.1; Sex + 1.6 varying room temperature + 1.8; stuffy “bad” air + 2.9; dust and dirt + 1.7; atopy + 2.1; light	6	0.44	0.07	−88.1
**SBS score** = 0.5 + 2.0; Sex + 1.7 varying room temperature + 1.8; stuffy “bad” air + 3.7; dust and dirt + 1.7; atopy + 3.9; workload	6	0.44	0.07	−88.1

Light (dim or causes glare and/or reflections).

**Table 5 ijerph-17-03635-t005:** Estimates for impact level in each dependent variable.

Risk Factors	Rank	LMG
Dust and dirt	1	0.10
Light (dim or causes glare and/or reflections)	2	0.07
Stuffy “bad” air	2	0.07
Dry air	3	0.06
Sex	3	0.06
Varying room temperature	4	0.05
Atopy	5	0.04
Workload	6	0.02

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
