# Peer review of "Working Conditions and Sick Building Syndrome among Health Care Workers in Vietnam"

_ijerph, 2020, doi:10.3390/ijerph17103635_

Round 1
Reviewer 1 Report
Sick building syndrome (SBS) is a global problem. The authors researched in Vietnam, but it can be concluded that the problem occurs in other countries.
The results of research refer to health care workers (HCW). The HCWs are expected to be healthy and to feel good during their unique work.
SBS is mostly present in buildings with bad ventilation and very airtight ones. Some health and feelnessles symptoms are connected with SBS.
I have a few concerns and suggestions:
1) In the introduction part, I would expect to get some information about exact symptoms that will be evaluated with the short description on which of SBS factors are connected with each symptom. SBS is a collection of certain factors that may have a definite impact on people's feelings. SBS itself is not a particular cause.
2) The research was done from March to June. Have people's responses regarding the sense of thermal comfort been assessed, taking into account different external conditions, or maybe in Vietnam during this period, the parameters of the outside air are constant?
3) Was the ventilation assessed in the rooms in which the surveyed people reside? Have feelings of comfort been associated with the type of ventilation? This seems important.
4) I would expect more information on the Indoor air quality in the building (temperature, relative humidity, CO2 concentration). Did you checked it?
5) SBS is strongly connected with the building (building envelope, ventilation type, furnishing). Can SBS be reduced by giving the knowledge of SBS to HCW? It was stated in the Conclusions. I think that this knowledge should be provided to architects, builders, owners, and maintenance of the building.
6) The number of people who took part in the survey is small, but it's probably hard to get a larger sample.
7) Conclusions are not very clear:
- providing knowledge to HCW will reduce SBS - How? Please elaborate,
- how would the findings contribute to SBS-prevention policy-making? Have HCWs have any influence in SBS? SBS is connected to the building and its equipment, IAQ, ventilation. I don't know how your findings could inspire HCWs to make SBS-prevention policy,
- could you give specific indicators on how to promote IAQ to prevent SBS?
Author Response
PASTEUR INSTITUTE
HO CHI MINH CITY
167 Pasteur St, Ward 8, District 3
Ho Chi Minh City, Vietnam
T+8428 3820-1419 F+8428 3820-352
E http://ww.pasteurhcm.gov.vn/
The Editor
International Journal of Environmental Research and Public Health
16 May 2020
Dear Sir/Madam,
RE: Submission of a manuscript
On behalf of co-authors, I would like to submit our manuscript entitled, “Working conditions and sick building syndrome among health care workers in Vietnam” to your journal and request that you consider publishing it as an article.
In this manuscript, we make a point-by-point response to the comments in the table below.
Thank you for your consideration of this manuscript.
Yours sincerely,
Dr. Cuong Hoang Quoc, M.D; Ph.D.
Corresponding Author
Direct Board, Pasteur Institute Ho Chi Minh City, Vietnam Ministry of Health
167 Pasteur Institute, Ward 8, District 3, Ho Chi Minh City, Vietnam
T: (+08) 933 918 918; E: [email protected]
No. |
Comments |
Replies |
Reviewer 1 |
||
|
In the introduction part, I would expect to get some information about exact symptoms that will be evaluated with the short description on which of SBS factors are connected with each symptom. SBS is a collection of certain factors that may have a definite impact on people's feelings. SBS itself is not a particular cause. |
Thanks for your recommendation. I have added some information below: For example, high CO2 levels related to symptoms like nausea, headaches, nasal irritation, dyspnea, and throat dryness, high light intensity were likely to report skin dryness, eye pain, and malaise. Similarly, high temperatures correlated with symptoms such as sneezing, skin redness, itchy eyes, and headache, while high relative humidity was likely to report sneezing, skin redness, and pain of the eyes. Lower respiratory symptoms were relationship with high workload, longer work hours, chemical exposure, migraine, and exposure to new interior painting.
|
|
The research was done from March to June. Have people's responses regarding the sense of thermal comfort been assessed, taking into account different external conditions, or maybe in Vietnam during this period, the parameters of the outside air are constant? |
Vietnam belongs to the tropical climate area; therefore, there was quite a difference between external conditions and internal conditions. However, participants always stay at the workplace as overload work and only shift from one department to another. In this study, we found that varying room temperature had significantly higher SBS score. |
|
Was the ventilation assessed in the rooms in which the surveyed people reside? Have feelings of comfort been associated with the type of ventilation? This seems important. |
Thank you very much for your comments. The study setting established in 2013 and met the international criteria, they had maintenance plans each year. In this study, we had only measured the airspeed indicator. A previous study also showed that ventilation associated with work productivity, there's a significant increase in production as ventilation rates increase, by 1.7% for every two-fold increase of ventilation rate. Therefore, we took the ventilation into a limitation in the discussion part. |
|
I would expect more information on the Indoor air quality in the building (temperature, relative humidity, CO2 concentration). Did you check it? |
Thanks for your suggestion. We had added these indicators to this manuscript. I hope that it is better than before. |
|
SBS is strongly connected with the building (building envelope, ventilation type, furnishing). Can SBS be reduced by giving the knowledge of SBS to HCW? It was stated in the Conclusions. I think that this knowledge should be provided to architects, builders, owners, and maintenance of the building. |
Thank you so much. We have already added it into conclusion. Taken together with our findings, the working condition criteria approach trained for architects, builders, owners, and maintenance of the building is highly recommended for indoor air quality improvement. Furthermore, larger-sample studies about physical stressors are urgent need to better manage SBS. |
|
The number of people who took part in the survey is small, but it's probably hard to get a larger sample. |
Yes. To be honest, HCWs were always busy so it is quite difficult to invite them to join the study. In this study, we recruited the number of participants as much as we can. |
|
7) Conclusions are not very clear: providing knowledge to HCW will reduce SBS - How? Please elaborate, how would the findings contribute to SBS-prevention policy-making? Have HCWs have any influence in SBS? SBS is connected to the building and its equipment, IAQ, ventilation. I don't know how your findings could inspire HCWs to make SBS-prevention policy, could you give specific indicators on how to promote IAQ to prevent SBS? |
Thank you so much. We have revised and rewritten the conclusion below: HCWs in the hospitals are exposed to the physical stressors in working conditions which have a potential increased SBS score and the severity of SBS. The risk factors could be varying room temperature, dust, and dirt, light, and stuffy “bad” air were the predominant influences for SBS. Our study reveals that working conditions are important and significantly associated with SBS. Taken together with our findings, the working condition criteria approach trained for architects, builders, owners, and maintenance of the building is highly recommended for indoor air quality improvement. Furthermore, larger-sample studies about working conditions are urgent need to better manage SBS. |

Reviewer 2 Report
Comments on manuscript ijerph-801485, “Predictors of sick building syndrome among health care workers in Vietnam”. The manuscript deals with an interesting topic and it is suitable for the journal. However, I believe it cannot be accepted in its present form.
The authors must attend the following comments
English of the manuscript should be thoroughly reviewed because there are many grammatical and spelling errors in the text.
In the abstract, the authors should mention the influencing factors they studied. It is essential that both the title and the abstract give an honest indication of what the paper contains. The abstract should be concise and precise, indicating to the potential
reader two things: (a) what was done, and (b) important results obtained.
In line 13, replace “interviews was conducted for collecting” by “interviews were conducted to collect”
In line 27, replace “was” by “is”
In line 28, write “for” between difficulty and concentrating
In line 29, replace “affected” by “affects”
In line 34, replace “was more pollutant” by “is sometimes more polluted”
In line 40, replace “was” by “is”
In line 41, it seems that “being female workers” does not correspond to this sentence because the authors are mentioning factors associated with SBS
In line 43, replace “was” by “is”´
In line 45 erase the word “present”
The contribution of the manuscript is not well presented. Probably they could write it as: This study aims to evaluate the correlation of SBS and the symptoms among HCW’s.
In line 58 replace “HCWs was needed” by “HCWs were needed”
In line 82 write “to” between “related” and “working”
In line 96 erase “is” after SBS
In line 97 write “for” between “difficulty” and “concentrating”
In line 128 write “they” between “and” and “had”,
It seems that the sentence of line 128 is incomplete, I believe they must write something after the words three years, please check.
About lines 138-139, how frequent were these symptoms felt?
Join tables of page 5 and 6 if they are the same table
Nothing is mentioned about the air conditioning systems, did the buildings they analyzed have air conditioners?
The conclusion are not meaningful, there is no connection to the Introduction by way of the question(s) or hypotheses you posed.
The references do not have the same format
Author Response
PASTEUR INSTITUTE
HO CHI MINH CITY
167 Pasteur St, Ward 8, District 3
Ho Chi Minh City, Vietnam
T+8428 3820-1419 F+8428 3820-352
E http://ww.pasteurhcm.gov.vn/
The Editor
International Journal of Environmental Research and Public Health
16 May 2020
Dear Sir/Madam,
RE: Submission of a manuscript
On behalf of co-authors, I would like to submit our manuscript entitled, “Working conditions and sick building syndrome among health care workers in Vietnam” to your journal and request that you consider publishing it as an article.
In this manuscript, we make a point-by-point response to the comments in the table below.
Thank you for your consideration of this manuscript.
Yours sincerely,
Dr. Cuong Hoang Quoc, M.D; Ph.D.
Corresponding Author
Direct Board, Pasteur Institute Ho Chi Minh City, Vietnam Ministry of Health
167 Pasteur Institute, Ward 8, District 3, Ho Chi Minh City, Vietnam
T: (+08) 933 918 918; E: [email protected]
No. |
Comments |
Replies |
Reviewer 2 |
||
|
English of the manuscript should be thoroughly reviewed because there are many grammatical and spelling errors in the text. |
Thank you very much for considering our manuscript. We apologized for my mistakes. We have already checked and hope that it will be better than previous version. |
|
In the abstract, the authors should mention the influencing factors they studied. It is essential that both the title and the abstract give an honest indication of what the paper contains. The abstract should be concise and precise, indicating to the potential |
We revised the title and abstract in detail below: Work environment and sick building syndrome among health care workers in Vietnam. Our study reveals that working conditions are important and significantly associated with SBS. Taken together with our findings, the physical stressors criteria approach trained for architects, builders, owners, and maintenance of the building is highly recommended for indoor air quality improvement. Furthermore, larger-sample studies about physical stressors are urgent need to better manage SBS |
|
In line 13, replace “interviews was conducted for collecting” by “interviews were conducted to collect” In line 27, replace “was” by “is” In line 28, write “for” between difficulty and concentrating In line 29, replace “affected” by “affects” In line 34, replace “was more pollutant” by “is sometimes more polluted” In line 40, replace “was” by “is” In line 41, it seems that “being female workers” does not correspond to this sentence because the authors are mentioning factors associated with SBS In line 43, replace “was” by “is”´ In line 45 erase the word “present” In line 58 replace “HCWs was needed” by “HCWs were needed” In line 82 write “to” between “related” and “working” In line 96 erase “is” after SBS In line 97 write “for” between “difficulty” and “concentrating” In line 128 write “they” between “and” and “had”, |
Thank you again for helping us revising line by line grammatical and spelling errors. We have revised and checked it again. |
|
About lines 138-139, how frequent were these symptoms felt? |
Thank you so much. We have added the frequent of feeling symptoms to manuscript. Participants weekly reported these symptoms. |
|
It seems that the sentence of line 128 is incomplete, I believe they must write something after the words three years, please check. |
It was replaced by “they had three years working experience. |
|
The contribution of the manuscript is not well presented. Probably they could write it as: This study aims to evaluate the correlation of SBS and the symptoms among HCW’s. |
We already changed as the sentence which you recommended: “This study aims to evaluate the correlation of SBS and the symptoms among HCWs”. |
|
Join tables of page 5 and 6 if they are the same table |
Yes, there are the same table. We tried my best to make it shorter than previous version. However, it also depends on format from the editors. |
|
The conclusion is not meaningful, there is no connection to the Introduction by way of the question(s) or hypotheses you posed. |
We have revised and rewritten in below: HCWs in the hospitals are exposed to the physical stressors in working conditions which have a potential increased SBS score and the severity of SBS. The risk factors could be varying room temperature, dust, and dirt, light, and stuffy “bad” air were the predominant influences for SBS. Our study reveals that working conditions are important and significantly associated with SBS. Taken together with our findings, the physical stressors criteria approach trained for architects, builders, owners, and maintenance of the building is highly recommended for indoor air quality improvement. Furthermore, larger-sample studies about physical stressors are urgent need to better manage SBS. |
|
The references do not have the same format |
We have checked and revised it based on the journal’s guideline |
|
Nothing is mentioned about the air conditioning systems, did the buildings they analyzed have air conditioners? |
Thanks for your comments. We have added several indicators including temperature, relative humidity, CO2, PM10, Airspeed, electromagnetic radiation, light, and level of noise. We hope that it is better than a previous version. |

Reviewer 3 Report
In the manuscript IJERPH-801485 authors present the results of a study aimed to assess correlates of Sick Building Syndrome (SBS) among Health-Care Workers (HCWs) in Vietnam. A total of 207 HCWs were recruited in a large-hospital-based cross-sectional survey; face to face interviews was conducted for collecting data on demographics, SBS-related symptoms, working environments, and conditions. Lindeman, Merenda, and Gold (LMG) test were used to investigate the predictors and its impact on the SBS. Results outlined that being female, atopy, varying temperature room, stuffy “bad” air dust and dirt had higher SBS scores. LMG test showed that dust and dirt, and stuffy “bad” air were the predominant risk factors for SBS. Authors found that SBS score and the severity of SBS were high among HCWs in Vietnam.
The study could be of interest, as it deals with a possible health and safety issue which could involve a category of workers. The adopted protocol is adequate to the purpose of the study. The introduction section provides enough elements to properly define the background and the problem statement. The paper is scientifically sound. In summary, this is a manuscript that that could be considered for the publication on IJERPH. Below a couple of minor comments that I ask you to consider and respond to.
Comments
- Please, consider to perform a linguistic and editorial revision of the text, which presents some errors and typos
- A part of the text is organized in very short chapters, which in some cases are little more than a list of parameters collected for the purpose of the study (2.2 - 2.5). Perhaps to facilitate reading it would be good to reorganize this part of the text and refer to the tables to list these parameters. in the same way paragraphs 3.1 to 3.5 are a few lines each. It may be more useful to organize them more efficiently. it's just general advice, though. I leave the decision to the authors.
- Just a consideration. If I have interpreted the study well, the questionnaire is based on the perception of the work environment by the subjects. Could this aspect have introduced a bias? (subjects who complain of SBS symptoms perceive the environment in a more negative way than others). Although difficult to undertake, the execution of inspections to confirm the information relating to the work environments obtained with the questionnaire, even only with the use of qualitative (checklist) or quantitative (physical, indoor air quality measurements) and the quality of the indoor environment) in the environments frequented by the subjects, it would have supported the evidence obtained. Can this be considered a limitation of the study?
Author Response
PASTEUR INSTITUTE
HO CHI MINH CITY
167 Pasteur St, Ward 8, District 3
Ho Chi Minh City, Vietnam
T+8428 3820-1419 F+8428 3820-352
E http://ww.pasteurhcm.gov.vn/
The Editor
International Journal of Environmental Research and Public Health
16 May 2020
Dear Sir/Madam,
RE: Submission of a manuscript
On behalf of co-authors, I would like to submit our manuscript entitled, “Working conditions and sick building syndrome among health care workers in Vietnam” to your journal and request that you consider publishing it as an article.
In this manuscript, we make a point-by-point response to the comments in the table below.
Thank you for your consideration of this manuscript.
Yours sincerely,
Dr. Cuong Hoang Quoc, M.D; Ph.D.
Corresponding Author
Direct Board, Pasteur Institute Ho Chi Minh City, Vietnam Ministry of Health
167 Pasteur Institute, Ward 8, District 3, Ho Chi Minh City, Vietnam
T: (+08) 933 918 918; E: [email protected]
Review 3 |
||
|
Please, consider to perform a linguistic and editorial revision of the text, which presents some errors and typos
|
Thank you so much for consideration our manuscript. We have checked and revised it line by line. |
|
A part of the text is organized in very short chapters, which in some cases are little more than a list of parameters collected for the purpose of the study (2.2 - 2.5). Perhaps to facilitate reading it would be good to reorganize this part of the text and refer to the tables to list these parameters. in the same way paragraphs 3.1 to 3.5 are a few lines each. It may be more useful to organize them more efficiently. it's just general advice, though. I leave the decision to the authors.
|
Thanks for your suggestions. Actually, we mean that we want to follow the questionnaire’s structure (demographics, working environment, working conditions, SBS score, environmental monitoring). we will try my best to organize it more efficiently.
|
|
Just a consideration. If I have interpreted the study well, the questionnaire is based on the perception of the work environment by the subjects. Could this aspect have introduced a bias? (subjects who complain of SBS symptoms perceive the environment in a more negative way than others). Although difficult to undertake, the execution of inspections to confirm the information relating to the work environments obtained with the questionnaire, even only with the use of qualitative (checklist) or quantitative (physical, indoor air quality measurements) and the quality of the indoor environment) in the environments frequented by the subjects, it would have supported the evidence obtained. Can this be considered a limitation of the study?
|
Thanks for your comments. We added several chemical monitoring indicators in this version. In addition, we also added some limitations to the study. We hope that this version is better than the previous version.
|

Round 2
Reviewer 1 Report
I am satisfied with all the changes that were provided by the authors. I believe that the manuscript can be accepted.